# A Deep Counterfactual Framework for High-Flow Nasal Cannula and Non-Invasive Ventilation Recommendations for Acute Respiratory Failure

Xiaolei Lu*, Michael Miller†, Alex K. Pearce†,
Kai Zheng‡, Atul Malhotra†, Shamim Nemati*

*Abstract*—High-flow nasal cannula (HFNC) and non-invasive ventilation (NIV) are commonly used respiratory support therapies for acute respiratory failure (ARF). However, current randomized trials provide limited guidance for individualized treatment decisions in this patient population. We propose RepFlow-CFR, a deep counterfactual inference model designed to estimate the individualized treatment effects (ITEs) of HFNC versus NIV. The model was applied to retrospective data from ICU cohorts at two independent health systems, UC San Diego (UCSD) Health and UC Irvine (UCI) Health. The primary outcome was the need for invasive mechanical ventilation (IMV). After adjusting for confounders, a multivariable logistic regression analysis at the UCSD site showed that concordance with the RepFlow-CFR model's recommendations was significantly associated with a lower risk of IMV. Specifically, the odds ratio (OR) for IMV was 0.661 ($p<0.001$) for concordance with a NIV recommendation and 0.677 ($p=0.019$) for concordance with an HFNC recommendation. These results demonstrated a more significant and consistent protective effect compared to baseline methods like Causal Forest and X-learner. The findings underscore the model's potential to provide data-driven, personalized guidance for respiratory support decisions in critically ill patients.

*Index Terms*—Counterfactual Inference, Individualized Treatment Effect, Respiratory Failure, High-Flow Nasal Cannula, Non-Invasive Ventilation

## I. INTRODUCTION

Acute respiratory failure (ARF) remains one of the most critical conditions requiring prompt intervention in intensive care units (ICUs). As the need for respiratory support escalates, two commonly adopted non-invasive modalities are high-flow nasal cannula (HFNC) and non-invasive ventilation (NIV) [1]–[6]. These therapies can help prevent deterioration requiring invasive mechanical ventilation (IMV), which is associated with higher risks of complications and mortality. However, selecting between HFNC and NIV is often challenging due to overlapping indications and patient heterogeneity [7]–[10].

Randomized controlled trials (RCTs) have assessed the comparative effectiveness of HFNC and NIV, but often provide inconsistent conclusions, primarily due to their focus on average treatment effects and narrowly defined populations. Clinical guidelines recommend specific modalities for well-characterized scenarios, such as NIV for hypercapnic COPD exacerbations or HFNC for de novo hypoxemia [9], [10]. Yet, many ICU patients do not fit neatly into these guideline-defined categories, and coexisting conditions often complicate the decision-making process. This variability underscores the need for individualized, data-driven guidance that can adapt to the nuances of each patient.

To address this, machine learning (ML) offers tools to estimate individualized treatment effects (ITEs) from observational data by leveraging high-dimensional clinical features [11]–[13]. However, many existing ML models are limited by confounding bias and loss of clinically important covariates during representation learning [14], [15]. In this study, we introduce RepFlow-CFR, a deep counterfactual inference model that combines counterfactual regression [14] with conditional normalizing flows [16] to estimate robust ITEs. By modeling both observed and latent confounding, our approach generates interpretable predictions of IMV risk under HFNC and NIV, aiming to improve decision support for personalized respiratory therapy.

## II. METHODS

### A. Study Design and Cohort

We conducted a retrospective study using de-identified electronic health records from adult patients ($\geq$ 18 years) admitted to ICUs at UC San Diego Health (UCSD) between January 1, 2016, and December 31, 2023, and at UC Irvine Health (UCI) between January 1, 2021, and August 31, 2024. Patients were included if they had an ICU stay of at least 5 hours, had recorded vitals and labs before prediction, and were not invasively mechanically ventilated before ICU admission. Each ICU stay was treated as a separate encounter. We excluded encounters with Do Not Resuscitate (DNR) orders and those involving surgery within 24 hours to avoid confounding from perioperative care. Ethical approval was obtained from the UC San Diego Institutional Review Board.

### B. Vent.io Risk Threshold and Treatment Definition

To identify patients at high risk for invasive mechanical ventilation (IMV), we used the pretrained Vent.io respiratory failure model [17]. The model outputs a risk score for IMV

*Department of Biomedical Informatics, University of California, San Diego, USA. Email: xil270@health.ucsd.edu, snemati@health.ucsd.edu

†Division of Pulmonary, Critical Care, and Sleep Medicine, University of California, San Diego, USA. Emails: mam038@health.ucsd.edu, apearce@health.ucsd.edu, amalhotra@health.ucsd.edu

‡Department of Informatics, University of California, Irvine, USA. Email: kai.zheng@uci.edu

based on a 5-class labeling scheme, and we applied a threshold corresponding to 60% sensitivity to define high risk. We defined *T0* as the first time the score crossed this threshold. We then identified patients who received HFNC or NIV as their first respiratory support after *T0*. This defined our early intervention cohort, referred to as HFNC/NIV group.

## C. Feature Processing

We extracted 50 vitals and labs, 6 demographic features, 12 features related to the components of the SOFA and SIRS criteria, 12 medication categories, and 62 comorbidities. Vitals/labs were resampled into hourly bins; multiple measurements per hour were aggregated using medians. For each variable, we computed three views: baseline (72-hour average), local trend (delta), the time since the variable was last measured (TSLM). Missing values were forward-filled up to 24 hours, with remaining gaps imputed using the training cohort median, which yielded 150 input features per encounter.

## D. RepFlow-CFR Framework

We proposed the RepFlow-CFR model, a flow-based confounder adjustment model that integrates representation learning, normalizing flows and counterfactual inference. Figure 2 presents the architecture overview of the RepFlow-CFR model, which contains three modules as follows.

*Stage 0 (CFR-based representation learning):* We utilized the counterfactual regression architecture (CFR) [14] that includes shared representation layers and two distinct heads for predicting outcomes under different treatments. The shared representation layers, based on the Vent.io architecture, include a TSLM layer for adjusting the importance of labs and vitals, followed by a feedforward neural network. By training the CFR model, the shared representation is encouraged to balance the distribution of measured confounders across treatment groups. The loss function is formulated as:

$$L_0 = \mathbb{E}[L_{\text{CFR}}(x, a, y) + \lambda \cdot \text{IPM}_G(\{\phi_i\}_{i,a=\text{HFNC}}, \{\phi_i\}_{i,a=\text{NIV}})],$$

where $\text{IPM}_G$ is the empirical probability metric (e.g., Wasserstein distance), $L_{\text{CFR}}$ is the prediction loss, and $\lambda$ denotes the trade-off parameter that balances prediction accuracy and representation distribution matching.

After training Stage 0, we assume that the learned representation $\phi$ captures sufficient information from measured covariates such that potential outcomes are conditionally independent of treatment assignment, given $\phi$ and a latent variable $u$ representing unmeasured confounding as:

$$Y(A) \perp\!\!\!\perp A \mid \phi, u,$$

where $Y$ and $A$ denote the outcome and treatment, respectively.

The observed distribution is defined as $p(y \mid \phi, a) = \int p(y \mid \phi, u, a) p(u \mid \phi, a) du$. The interventional distribution, which removes the influence of $a$ on $u$ is

$$p(Y(a) = y \mid \phi) = \int p(y \mid \phi, u, a) p(u \mid \phi) du.$$

If $p(u \mid \phi, a) = p(u \mid \phi)$, we will have $p(Y(a) = y \mid \phi) = p(y \mid \phi, a)$. However, in observational studies, this assumption rarely holds due to treatment assignment bias or unmeasured confounding. Directly using $p(y \mid \phi, a)$ to estimate counterfactual outcomes would lead to biased inference. To address this, RepFlow-CFR includes two additional stages to explicitly model and account for this hidden bias.

*Stage 1 (Modeling outcome distribution):* We used a conditional normalizing flow (CNF) to model the observed outcome distribution $p(y \mid \phi, a)$, where $a \in \{\text{NIV}, \text{HFNC}\}$. The CNF learns an invertible transformation $f_{\phi,a}^1$ that maps a standard normal latent variable $U \sim \mathcal{N}(0, I)$ to the outcome space. The loss function is formulated as:

$$L_1 = \sum_{i=1}^n -\log p(f_{\phi_i, a_i}^1(U) = y_i).$$

*Stage 2 (Adjusting for hidden confounding):* Since the learned representation $\phi$ may not satisfy unconfoundedness due to unmeasured confounding, we introduced a second CNF $f_{\phi,a}^2$. It transforms a new latent variable $\tilde{U} \sim \mathcal{N}(0, I)$ to an interventional latent variable that approximates $p(u \mid \phi)$. The resulting latent sample is then passed through the Stage 1 transformation $f_{\phi,a}^1$, which shifts the latent distribution $p(u \mid \phi, a)$ toward the interventional distribution $p(u \mid \phi)$.

This adjustment allows us to account for hidden biases arising from factors like clinician decision-making, treatment selection bias, and unobserved patient severity. The loss function is formulated as:

$$L_2 = \sum_{i=1}^n -\log p(f_{\phi_i, a_i}^1(f_{\phi_i, a_i}^2(\tilde{U})) = y_i).$$

## E. Training and Evaluation

The UCSD dataset was split into 80% training and 20% validation. Each model stage was trained independently using the Adam optimizer with early stopping based on validation loss. Bayesian optimization tuned learning rates, flow depth, and regularization strength. To ensure robustness of model estimation during inference, we drew 100 latent samples per encounter from the Stage 2 CNF and mapped them through the Stage 1 transformation to obtain predicted outcomes. These were then averaged to produce stable estimates for each treatment condition. Model performance, including both predictive accuracy and ITE estimation quality, was evaluated on the entire UCSD early HFNC/NIV cohort. To assess generalizability, we further conducted external validation on an independent early HFNC/NIV cohort from UCI site.

The predicted Individual Treatment Effect (ITE) was defined as the difference between the predicted probability of IMV under NIV and under HFNC, when each was given as the first intervention following the Vent.io T0 timepoint. To assess predictive performance, we reported two standard metrics for IMV prediction: the Area Under the Receiver Operating Characteristic Curve (AUC) and the Area Under the Precision-Recall Curve (PR-AUC). ITE estimation quality was further evaluated by examining patient outcomes under

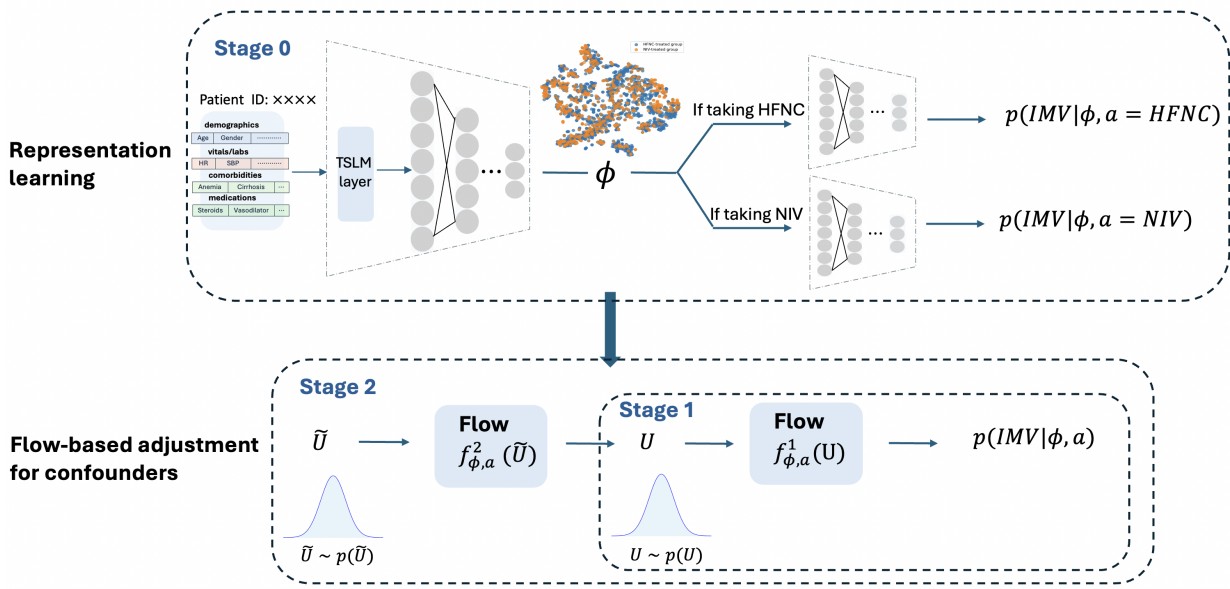

Fig. 1. Overview of the RepFlow-CFR model, which contains CFR-based representation learning, output distribution modeling and hidden confounding adjustment.

treatment concordance. As baseline comparisons, we included commonly used data-driven ITE estimation methods in clinical settings, including Causal Forest [11], X-Learner [12], and the CFR model.

## III. RESULTS

### A. Cohort Characteristics

We identified 1,956 ICU encounters from UCSD and 169 from UCI that met inclusion criteria. Patients were stratified based on the initial post-T0 treatment: HFNC or NIV. Compared to HFNC-treated patients, those receiving NIV had higher comorbidity burden and were more likely to have chronic pulmonary or cardiovascular conditions.

### B. IMV Predictive Performance

RepFlow-CFR achieved an AUC of $0.820$ and a PR-AUC of $0.566$ on the UCSD early HFNC/NIV cohort, which is comparable to the baseline CFR model (AUC: $0.821$, PR-AUC: $0.571$). However, when externally validated on the UCI early HFNC/NIV cohort, performance declined, with RepFlow-CFR achieving an AUC of $0.630$ and a PR-AUC of $0.444$, compared to CFR's AUC of $0.656$ and PR-AUC of $0.415$. To improve generalizability, we fine-tuned both CFR and RepFlow-CFR using 25% of the UCI early HFNC/NIV cohort. After fine-tuning, RepFlow-CFR achieved an AUC of $0.727$ and a PR-AUC of $0.553$, while CFR achieved an AUC of $0.758$ and a PR-AUC of $0.590$ on the UCI site.

### C. Outcomes with Treatment Concordance

Our primary clinical outcome was the need for invasive mechanical ventilation (IMV). We evaluated whether patients received treatments in concordance with the ITE predicted preferred treatment: NIV or HFNC, and compared IMV outcomes across different ITE estimation methods. Treatment concordance was defined as receiving the treatment predicted to be beneficial (concordant), while discordance referred to receiving the opposite treatment.

As illustrated in Figure 2, IMV rates were generally lower in the concordant group than in the discordant group across both sites, particularly for patients predicted to benefit from NIV. At UCSD, all methods showed lower IMV rates in the NIV-concordant group, with RepFlow-CFR achieving a rate of 19.27% compared to 27.06% in the discordant group. For HFNC, RepFlow-CFR also demonstrated the most favorable concordance effect, with a concordant rate of 20.07% versus 25.93% discordant. At UCI with RepFlow-CFR HFNC-concordant patients had an IMV rate of 13.33%, substantially lower than the 20.00% observed in the discordant group. While most methods showed consistent trends favoring concordance, X-learner at UCI showed a slightly higher IMV rate in the HFNC-concordant group (28.00%) than discordant (18.75%), deviating from the overall pattern.

To evaluate whether concordance with the predicted treatment is independently associated with reduced need for IMV, we conducted multivariable logistic regression adjusted for age, gender, SOFA score, CCI score, and Vent.io score (Table II). The strongest effect was observed for RepFlow-CFR at the UCSD site, where concordance with both NIV and HFNC was significantly associated with lower IMV risk (OR=0.661, $p < 0.001$ and OR=0.677, $p = 0.019$, respectively). In contrast, baseline models such as Causal Forest and X-learner did not consistently show significant protective effects. For example, the CFR model at UCSD showed that HFNC concordance was paradoxically associated with increased IMV risk (OR=1.846, $p < 0.001$), highlighting the potential harm

TABLE I
BASELINE CHARACTERISTICS OF PATIENTS IN UCSD AND UCI EARLY HFNC/NIV COHORTS.

| Variable | UCSD (NIV) | UCSD (HFNC) | UCI (NIV) | UCI (HFNC) |
|---|---|---|---|---|
| **Characteristic** | | | | |
| Encounters, N | 591 | 1365 | 38 | 131 |
| Age (years), mean (SD) | 63 (16.3) | 61 (16.6) | 66 (15.0) | 63 (17.6) |
| Male, N (%) | 369 (62.4) | 806 (59.0) | 20 (52.6) | 86 (65.6) |
| **Organ dysfunction, median (IQR)** | | | | |
| Charlson Comorbidity Index | 3.0 (1.0–5.0) | 2.0 (1.0–5.0) | 3.0 (1.0–6.0) | 2.0 (0.0–3.0) |
| Congestive Heart Failure, N (%) | 245 (41.5) | 315 (23.1) | 15 (39.5) | 29 (22.1) |
| Chronic Obstructive Pulmonary Disease, N (%) | 161 (27.2) | 234 (17.1) | 4 (10.5) | 8 (6.1) |
| SOFA score at Vent.io T0, median (IQR)[a] | 1.0 (0.0–3.0) | 1.0 (0.0–3.0) | 2.0 (1.0–4.0) | 1.0 (0.0–3.0) |
| **Outcomes, N (%)** | | | | |
| IMV[b] | 122 (20.6) | 334 (24.5) | 9 (23.7) | 38 (29.0) |
| Mortality | 147 (24.9) | 431 (31.6) | 9 (23.7) | 29 (22.1) |
| Hospice | 6 (1.0) | 11 (0.8) | 5 (13.2) | 31 (23.7) |

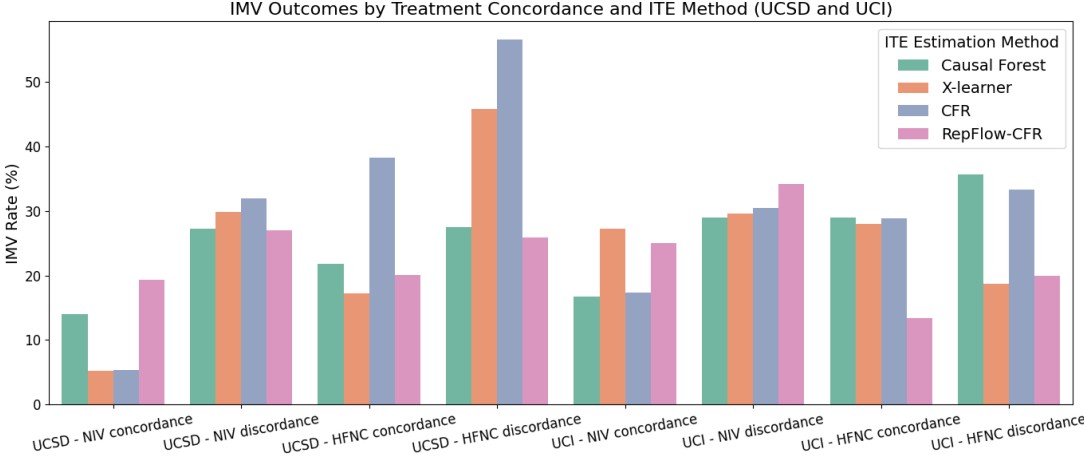

Fig. 2. Comparison of IMV outcomes by treatment concordance and ITE estimation method. Bars represent IMV rates across four treatment groups (NIV concordance, NIV discordance, HFNC concordance, HFNC discordance) for each ITE method.

of following incorrect recommendations.

## IV. DISCUSSION

We developed and validated RepFlow-CFR, a deep counterfactual model designed to estimate ITEs of HFNC versus NIV as the initial respiratory support for critically-ill ICU patients at risk of invasive mechanical ventilation. The model integrates deep representation learning and conditional normalizing flows to account for both measured and unmeasured confounders. Using 100 latent samples per encounter, we conducted a sensitivity analysis to estimate counterfactual outcomes under both treatment arms. Patients whose treatments aligned with model-predicted ITEs had significantly lower rates of IMV in both development and external validation cohorts. Compared to baseline methods including Causal Forest, X-Learner, and conventional CFR, RepFlow-CFR demonstrated more robust and consistent performance, particularly in external settings after fine-tuning.

These findings emphasize the limitations of relying solely on average treatment effects from RCTs, which may mask meaningful heterogeneity in individual patient responses. Although clinical guidelines offer recommendations for specific subpopulations (e.g., those with respiratory acidosis from COPD or cardiogenic pulmonary edema), many ICU patients present with complex, overlapping conditions not addressed by such guidelines. Consequently, treatment decisions are often based on clinical judgment, which may vary across providers. RepFlow-CFR offers a data-driven complement to existing guidelines, particularly when ambiguity exists. Its ability to model nuanced patient characteristics and predict individualized outcomes suggests potential utility in guiding escalation respiratory support decisions in critical care settings.

Nonetheless, this study has important limitations. As a retrospective analysis, it remains subject to the inherent constraints of observational data, including potential biases from undocumented clinical indicators or clinician intent. Although our CNF-based sensitivity analysis mitigates some of these concerns, prospective validation is needed to confirm clinical effectiveness. Moreover, our cohorts were drawn from two academic hospitals in the same geographic region, which may limit generalizability to broader or non-academic populations. Future research should evaluate the integration of RepFlow-CFR into clinical workflows, ideally combining model-predicted ITEs with clinician expertise and existing

TABLE II

MULTIVARIABLE LOGISTIC REGRESSION RESULTS (ODDS RATIOS AND p-VALUES) FOR PREDICTING THE NEED FOR IMV

| Method | Site | NIV concordance | HFNC concordance | Age | Gender | CCI | SOFA | Vent.io |
|---|---|---|---|---|---|---|---|---|
| Causal Forest | UCSD | **0.443** ($p < .001$) | **0.778** ($p = .033$) | 0.988 | 1.046 | 0.929 | 1.044 | 1.276 |
| | UCI | 0.415 ($p = .153$) | 0.903 ($p = .786$) | 0.994 | 1.046 | 1.089 | 1.051 | 1.032 |
| X-learner | UCSD | **0.109** ($p < .001$) | **0.356** ($p < .001$) | 0.984 | 1.035 | 0.932 | 1.066 | 1.267 |
| | UCI | 0.409 ($p = .089$) | 0.731 ($p = .397$) | 1.012 | 0.723 | 0.971 | 1.094 | 1.292 |
| CFR | UCSD | **0.185** ($p < .001$) | **1.846** ($p < .001$) | 0.992 | 1.039 | 1.049 | 1.120 | 1.294 |
| | UCI | 0.495 ($p = .245$) | 0.987 ($p = .783$) | 0.992 | 1.042 | 1.069 | 1.038 | 0.972 |
| RepFlow-CFR | UCSD | **0.661** ($p < .001$) | **0.677** ($p = .019$) | 0.988 | 1.003 | 0.933 | 1.047 | 1.143 |
| | UCI | 0.482 ($p = .373$) | 0.243 ($p = .121$) | 0.988 | 0.938 | 0.988 | 1.092 | 0.747 |

protocols to build interpretable, actionable decision-support tools.

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
