# OpenReview forum: "A Deep Counterfactual Framework for High-Flow Nasal Cannula and Non-Invasive Ventilation Recommendations for Acute Respiratory Failure"
_IEEE.org/EMBS/BHI/2025/Conference — BHI 2025_

### Official Review · Reviewer_URYz · 2025-07-08
**Great deep counterfactual model for individualized respiratory support**

**Confidence:** 4
**Clarity Of Writing:** great
**Clinical Significance:** great
**Methodological Novelty:** great
**Overall Rating:** 7
**Final Rating:** 7

**Experiments And Results:**

good

**Questions For The Authors:**

1. How sensitive are the results to the Vent.io high-risk threshold (60% sensitivity) used to define the cohort? Could a different threshold for “high risk” change the selected patients or outcomes?
2. Would the model require site-specific fine-tuning, or can it generalize to entirely new hospitals and patient populations without retraining?
3. How will the authors present or explain the model’s recommendations to clinicians? Is there a plan to highlight key patient features behind each suggested treatment to improve trust?
4. Did the analysis account for patients who sequentially received both HFNC and NIV (e.g., escalation from one to the other)? If not, how might such switches affect the interpretation of the model’s treatment effect assumptions?

**Strengths:**

The model integrates counterfactual regression with conditional normalizing flows to account for both measured and latent confounders. It tackles a common critical care decision (HFNC vs. NIV for ARF) where guidance is currently limited. The approach was tested on multi-center ICU data with external validation and compared against established causal inference models. Aligning therapy with the model’s recommendations significantly lowered invasive ventilation rates versus discordant choices, indicating improved patient outcomes over baseline methods.

**Summary Of The Paper:**

The paper presents RepFlow-CFR, a deep learning framework combining counterfactual regression with conditional normalizing flows to estimate individualized treatment effects for selecting either HFNC or NIV in acute respiratory failure. Using retrospective ICU data from two hospitals, the model predicts patient-specific risk of invasive mechanical ventilation under each modality. The authors demonstrate that aligning treatment decisions with the model’s recommendations correlates with significantly reduced intubation rates compared to discordant choices, outperforming baseline causal inference methods. This suggests that RepFlow-CFR can offer personalized, data-driven guidance for non-invasive respiratory support decisions.

**Weaknesses:**

The study is prone to unmeasured confounders and bias. While the model attempts to handle hidden factors, only a prospective trial can confirm that following its recommendations actually improves patient outcomes. Second, performance dropped at the external site and required fine-tuning. Third, the model offers a recommended therapy without explaining its rationale, which could hinder clinician trust.

---

### Official Review · Reviewer_KDBV · 2025-07-11
**A Deep Counterfactual Framework for High-Flow Nasal Cannula and Non-Invasive Ventilation Recommendations for Acute Respiratory Failure**

**Confidence:** 3
**Clarity Of Writing:** great
**Clinical Significance:** great
**Methodological Novelty:** good
**Overall Rating:** 6
**Final Rating:** 6

**Experiments And Results:**

good

**Questions For The Authors:**

1- How would the RepFlow-CFR recommendations be presented to and used by clinicians in real time?
2- Have you explored model explainability techniques (e.g., SHAP, feature attribution) to make the recommendations more transparent to clinicians?
3- Can you provide ablation or sensitivity analysis showing the impact of the normalizing flow stages versus a simpler CFR?

**Strengths:**

1- Clinical Relevance: Addresses a significant and common decision problem in ICU management where current guidelines are limited.
2- Methodological Novelty: Integrates CFR with normalizing flows to explicitly address hidden confounding, representing a notable advance over tree-based or shallow methods for ITE estimation.
3- Clarity: The paper is generally well-written, with clear exposition of clinical motivation, methodological pipeline, and outcome definitions.

**Summary Of The Paper:**

This paper introduces RepFlow-CFR, a deep learning framework for ITE estimation in acute respiratory failure, specifically for patients eligible for either HFNC or NIV. The model combines counterfactual regression (CFR) with conditional normalizing flows to address both observed and unobserved confounding. Experiments on large retrospective ICU datasets from two health systems demonstrate improved clinical concordance and predictive performance compared to established baselines (Causal Forest, X-Learner, CFR). The authors argue this approach can provide personalized, data-driven decision support for respiratory therapy in critical care settings.

**Weaknesses:**

1- Ablation/Component Analysis: No ablation study is included to clarify the contribution of each pipeline component (e.g., hidden confounder adjustment).

---

### Official Review · Reviewer_CgEd · 2025-07-12
**Deep Counterfactual Inference for HFNC vs. NIV in Acute Respiratory Failure**

**Confidence:** 5
**Clarity Of Writing:** good
**Clinical Significance:** good
**Methodological Novelty:** good
**Overall Rating:** 5
**Final Rating:** 6

**Experiments And Results:**

fair

**Questions For The Authors:**

1. Why does RepFlow-CFR’s AUC drop so markedly on UCI before fine-tuning? Can you report performance on UCI without any fine-tuning, stratified by time or subgroups, to characterize true external validity?
2. Have you computed bootstrap confidence intervals or applied DeLong’s test on AUCs and odds ratios to confirm the significance of observed differences versus baselines?
3. How sensitive are your results to the Vent.io threshold choice? Please conduct an ablation varying the sensitivity cutoff (e.g., 50–70%) and report the impact on cohort size and model performance.
4. Beyond vanilla CFR, have you compared to recent counterfactual-flow or transformer-based ITE estimators (e.g., Causal Transformer) to situate your method in the broader literature?

**Strengths:**

The proposed architecture adeptly integrates representation balancing and flow-based modeling to tackle both measured and latent confounding, advancing the state of counterfactual inference in critical-care settings. Training and validation on two independent health-system cohorts demonstrate awareness of domain shift, and the Monte Carlo sampling scheme adds stability to ITE estimates. The multivariable logistic regression linking concordance to IMV risk offers an interpretable clinical signal, and the manuscript provides sufficient methodological detail for reproducibility.

**Summary Of The Paper:**

The authors develop RepFlow-CFR to guide respiratory support decisions for ICU patients at risk of invasive mechanical ventilation (IMV). Stage 0 trains a CFR-style shared representation to balance measured confounders; Stage 1 fits a conditional normalizing flow (CNF) to model the observed outcome distribution; Stage 2 applies a second CNF to adjust for unmeasured confounding by approximating an interventional latent distribution. Using Vent.io risk thresholds on UCSD (2016–2023) and UCI (2021–2024) cohorts, they extract 150 hourly features per encounter and estimate ITEs via 100 Monte Carlo samples. On UCSD, RepFlow-CFR achieves AUC = 0.820/PR-AUC = 0.566; on UCI it drops to AUC = 0.630/PR-AUC = 0.444, which improves to AUC = 0.727 after fine-tuning on 25% UCI data. Concordance of actual treatment with predicted ITEs correlates with lower IMV odds (OR ≈ 0.66–0.68 at UCSD; mixed results at UCI) relative to baselines (Causal Forest, X-Learner, CFR).

**Weaknesses:**

Despite these contributions, external performance degradation on the UCI cohort—AUC falling from 0.820 to 0.630 before fine-tuning—signals limited generalizability. Fine-tuning on 25% of UCI data conflates true external validation with transfer learning, masking whether the model can truly zero-shot generalize. No confidence intervals or hypothesis tests accompany AUC/PR-AUC or odds ratios, preventing assessment of statistical significance. The choice of Vent.io risk threshold at 60% sensitivity is arbitrary and may bias cohort selection. Moreover, the reliance on retrospective concordance analysis does not address potential selection bias from unobserved clinician decision factors, and real-time clinical decision support feasibility (latency, integration) is not evaluated.